# Pharmacotherapy problems and associated factors among type 2 adult diabetic patients on follow up at Mizan-Tepi University Teaching Hospital, Southwest Ethiopia

**Semere Welday Kahssay** (ORCID) *****, **Nebeyi Fisseha Demeke**

Department of Pharmaceutical Chemistry and Pharmacology, School of Pharmacy, College of Medicine and Health Sciences, Mizan-Tepi University, Mizan-Aman, Ethiopia

***** semere0409@gmail.com

**Data Availability Statement:** All relevant data are within the paper and its Supporting information files.

## Abstract

### Background

Over the past few decades, drug therapy problems (DTPs) have become a significant public health concern worldwide. DTPs in patients with diabetes are responsible for uncontrolled glycemia, disease worsening, early development of complications, high healthcare expenses, prolonged and recurrent hospitalizations, and mortality.

### Objectives

This study aimed to identify the prevalence of drug therapy problems and determine the associated factors among patients with type II Diabetes Mellitus at a University Teaching Hospital in Southwest Ethiopia.

### Methodology

Hospital-based cross-sectional study was conducted between September and October 2022. Data were collected through medical record reviews and interviewer-administered structured questionnaires, which were then analyzed using SPSS version 26. Cipolle's method was adapted for classification of DTPs. Bivariate followed by multivariate logistic regression analysis was used to assess the association between predictor variables and the outcome variable. P-value ≤ 0.05 was employed as a cut-off point to determine statistical significance.

### Result

Among 117 participants, 172 drug therapy problems (DTPs) were identified, with an average of 1.47 DTPs per patient, and 83 (70.9%) participants had at least one type of drug therapy problem. Of the seven DTPs identified, need additional drug therapy was the most common, 50 (42.7%), followed by non-compliance with medication, 45 (38.5%) and ineffective drug therapy, 25 (21.4%). Occupational status and comorbidity were factors that associated with the occurrence of DTPs. Farmers were approximately four times more likely to develop

**Funding:** The author(s) received no specific funding for this work.

**Competing interests:** The authors have declared that no competing interests exist.

DTPs than housewives were (adjusted odds ratio (AOR) = 3.56, 95% CI: 1.12–11.38, P = 0.03). The odds of drug therapy problems were twice as high in those with four comorbid conditions than in those without comorbidities (AOR = 1.95, 95% CI: 0.90–3.76, p = 0.02).

## Conclusion

In the current study, the proportion of type 2 diabetes patients with drug therapy problems was high. This potentially lead to uncontrolled glycemia and early development of comorbid conditions, increasing morbidity and mortality rates. This could be attributed to the failure to effectively integrate clinical pharmacy services in different hospital wards, which is the case in virtually all hospitals in Ethiopia.

## Introduction

Diabetes Mellitus (DM) is a common metabolic disease characterized by chronic hyperglycemia resulting from defects in insulin secretion, action or both. Type 1 DM which accounts for 5–10% of all diabetic cases is characterized by absolute insulin deficiency. Whereas, Type 2 diabetes is mainly due to insulin resistance with a relative insulin deficiency, and accounts for 90–95% of all diabetes cases. Gestational diabetes mellitus and other specific type (monogenic diabetes) are other forms of diabetes mellitus [1].

A recent International Diabetic Federation (IDF) report revealed that globally, approximately 537 million people between the ages of 20 and 79 years old are living with diabetes mellitus, with a global prevalence rate of 10.5%, which is expected to exceed 12% by 2045. Disturbingly, three out of four patients with diabetes live in low- and middle-income countries. Between 2021 and 2045, the number of individuals with diabetes mellitus in Africa, South-East Asia, and Europe is predicted to rise by 134%, 68%, and 13%, respectively. In 2021, approximately 6.7 million deaths and $966 billion global health expenditures were attributed to diabetes mellitus [2–4].

The crucial parts of diabetes treatment include diabetes self-management education, lifestyle modifications, and pharmacologic management of diabetes and comorbid disease conditions, such as hypertension and hyperlipidemia [5, 6]. Despite the type of treatment approach used, it is challenging to reach the desired blood sugar level [7]. According to some studies, approximately two-thirds of T2DM patients with hypertension fail to reach the desired blood pressure and sugar levels, primarily as a result of the emergence of drug therapy problems (DTPs) [8].

The presence of comorbid condition/s, age, income, and number of prescribed medications are some of the factors reported to contribute to the occurrence of DTPs in patients with T2DM [9–14]. Hypertension is the most common co-morbidity and can occur in up to 82% of patients with type 2 diabetes. The presence of comorbid condition increases the number of medications prescribed to patients with diabetes, which raises their risk of developing DTPs [15].

Drug therapy problems (DTPs) in patients with diabetes are responsible for uncontrolled glycemia, disease worsening, early development of complications, recurrent and prolonged hospitalizations, high healthcare costs, and mortality. Studies in some parts of Ethiopia have revealed that 62.4%–88% of patients with type 2 diabetes experience at least one type of DTP [11, 16]. Despite the severity of the problem, only a few studies have been conducted in Ethiopia, and no data are currently available for the southwest region. Thus, this study aimed to

assess drug therapy problems and associated factors among patients with type 2 diabetes at Mizan-Tepi University Teaching Hospital, Southwest Ethiopia.

## Methodology

### Study area and period

The study was conducted at Mizan-Tepi University Teaching Hospital, which is found in Mizan-Aman town, Bench- Sheko zone, Southwest Ethiopia peoples' region, and is around 569 km away from the capital city of Ethiopia, Addis Ababa, and 123 km from Bonga (the capital city of the region). MTUTH was built in 1981. The hospital serves more than 2 million people annually. In addition, the hospital provides services to people from Gambella Region, Sheka, and Kaffa Zones. The hospital has four central wards: Medical, Surgical, Pediatric, and Gynecology/Obstetrics. This study was conducted between September and October 2022.

### Study design

Hospital-based cross-sectional study was conducted.

### Populations

**Source populations.** All type 2 adult diabetic patients who visited the MTUTH diabetes outpatient clinic during the study period.

**Study population.** All adult patients with type 2 diabetes who fulfilled the inclusion criteria were included in this study.

**Inclusion and exclusion criteria.** *Inclusion criteria.* All adult patients with type 2 diabetes who had been on follow-up for at least the last three months at MTUTH and had received at least one antidiabetic drug.

*Exclusion criteria.* Severely ill patients who were unable to respond to the questions because of their medical condition, those with incomplete medical cards, and those who were unwilling to participate in the study were excluded.

### Sample size determination and sampling technique

The sample size was determined using a single population proportion formula, considering the following assumptions: the sample proportion was taken from the prevalence of DTP in a similar study (83.1%) [17] 95% confidence level and 5% marginal error.

$$N = \frac{Z_{a/2}*P*(1-P)}{d^2}$$

**Where** n = sample size

**Z** = is the level of confidence = 1.96 with 95% CI.

**P** = the prevalence of DTP = 0.831 (83.1%).

**d** = 0.05 margin of error (5%).

n = {(1.96)$^2$ (0.831) (1–0.831)}/ (0.05)$^2$ = 226

As the total number of patients with type 2 diabetes on follow-up at the MTUTH were less than 10,000, correction/reduction formula was used to obtain the minimum possible sample size.

$$N_f = \frac{n}{1 + \frac{n}{N}} = \frac{226}{1 + \frac{226}{200}} = 106$$

Where n = first calculated sample size

N = size of the source population (200)

$N_f$ = the minimum sample size

Adding 10% contingency (106*0.1 = 11) for the non-response rate yielded a final sample size of 117.

Consecutive/simple random sampling techniques was used to select the participants.

## Operational definition

**Drug therapy problem.** Any unfavorable event a patient encounters that is believed to be related to pharmacological therapy and prevents them from attaining the intended goals of therapy [16].

**Dosage too high.** If the patient was taking a drug at a higher dose than the recommended dose according to the standard treatment guidelines [18].

**Dosage too low.** If the patient was taking a drug at a lower dose than the recommended dose according to the standard treatment guidelines [18].

**Unnecessary drug therapy.** Unwanted addition of medication without indication for adding [19].

**Needs additional drug therapy.** If a patient is not receiving enough medication and has blood sugar levels outside of the acceptable range [19].

**Adverse drug reaction.** A proper drug dose producing unpleasant or severe side effects [19].

**Ineffective drug therapy.** Choosing inappropriate medications as per the accepted treatment protocols [18].

**Noncompliance.** If a patient scores more than three in MMAS-8 ((Morisky Medication Adherence Scale 8 item) for reasons such as misunderstanding of instructions, the patient prefers not to take the medication, the patient cannot properly take the drug, it is too pricey, and not readily available to the patient [20].

## Data collection method

**Data collection instrument and process.** A comprehensive structured questionnaire was prepared based on previous studies with minor modifications [17, 19, 21, 22]. It contained sociodemographic characteristics, medication and clinical related information, and possible DTP-contributing factors (S1 File). Minor modifications were made on the socio-demographic characteristics of the study subjects, in which some essential variables like social drug use and BMI (body mass index) were added. The collection of sociodemographic characteristics of the study participants and assessment of unreported adverse drug reactions were carried out during the patients' hospital visit, while clinical and medication-related information were collected retrospectively from the patients' medical records. The study participants were interviewed before they enter into the diabetes outpatient clinic. DTPs were evaluated and classified using the Cipolle's DTP identification tool (Online Appendix I) [23], which categorizes drug therapy problems into seven categories: dose too low, dose too high, need additional drug therapy, ineffective drug therapy, unnecessary drug therapy, non-compliance, and adverse drug reaction. Two pharmacists collected the data and assessed DTPs, and the possible causes of DTPs were identified from patients' medical records with reference to accepted practices.

**Data quality control.** A pre-test was conducted on 5% of the study population a week before the data collection period, and amendments were made accordingly. Training was provided to the data collectors mainly on the data collection tool, and close supervision was performed daily by the principal investigator. To verify the accuracy of the data, each data

collection process ended with a review of the completed questionnaires and recorded data. Any inaccuracies were corrected right away.

**Data processing and analysis.** The data were entered and analyzed using SPSS version 26. Descriptive statistics, including mean and standard deviation for continuous variables, and frequency and percentage for categorical data, were used to summarize sociodemographic and relevant clinical data. Bivariate and multivariate logistic regression analyses were performed to investigate the association between variables and the occurrence of DTPs. The variables were checked for collinearity. Variables with a p-value $\leq 0.25$ in the bivariate analyses, were further analyzed using multivariate logistic regression to control for the effect of confounders. In multivariate analysis, statistical significance was set at $p \leq 0.05$.

## Ethical considerations

Ethical clearance was obtained from Mizan-Tepi University, College of Medicine and Health Sciences, Research and Ethical Review Committee (CMHS/01621). Before starting data collection, permission was obtained from the study site department. The right of the participants to withdraw at any time or not to participate in the study was respected. Before conducting the interviews, the purpose of the study was explained to each participant and oral informed consent was obtained. Anonymity and confidentiality were maintained by removing identifiers and restricting access to the data. The study complied with the principles of the Declaration of Helsinki.

## Result

### Socio-demographic characteristics of the study participants

A total of 117 study participants participated in the study, giving a 100% response rate. Approximately two-thirds (66.7%) of the study participants were male. The mean age of the participants was 57.07 ± 13.13 years, with a range of 30–96 years. Nearly three-fourths (72.6%) of the patients with type 2 Diabetes Mellitus (T2DM) were overweight, and 6% were obese. Participants with the age range of –45–59 made up the most significant proportion (41.9%). 28 (23.9%) study participants had a family history of DM. Regarding marital status, almost all of the participants (97.4%) were married. About half of the participants (50.4%) had completed primary education, and 16.2% had higher education. The majority of the participants, 44 (37.6%), were merchants. Individuals from the Amhara, followed by those of Bench ethnicity, comprised 38.5% and 15.4% of the participants, respectively. In contrast, 88% of the participants had no history of social drug use (Table 1).

### Drug therapy problems (DTPs) among the participants

Of 117 adult T2DM patients on antidiabetic medications, 172 drug therapy problems were identified, with an average of 1.47 DTPs per patient. Among the study participants, 83 (70.9%) had at least one type of drug therapy problem. Of all the DTPs identified, 50 (42.7%) required additional drug therapy, followed by non-compliance to medication 45 (38.5%), made up the most significant proportion. The remaining unnecessary drug therapy, ineffective drug therapy, dose too low, adverse drug reactions, and dose too high were observed in 18 (15.4%), 25 (21.4%), 13 (11.1%), 5 (4.3%), and 16 (13.7%) of the participants, respectively (Table 2).

### Clinical and drug therapy conditions of adult patients with type 2 diabetes

As shown in Table 3, 78.6% and 88% of the participants had no evidence of renal and liver function tests, respectively. Among the study subjects, 27 (23.1%)developed microvascular

**Table 1. Socio-demographic characteristics of adult T2DM patients on follow up at MTUTH, 2022.**

| Variables | Category | Frequency | Percent (%) |
|---|---|---|---|
| Sex | Male | 78 | 66.7 |
| | Female | 39 | 33.3 |
| Age | ≤44 | 34 | 29.1 |
| | 45–59 | 49 | 41.9 |
| | ≥60 | 34 | 29.1 |
| Family history of DM | Yes | 28 | 23.9 |
| | No | 89 | 76.1 |
| BMI | ≤24.9 | 25 | 21.4 |
| | 25–29.9 | 85 | 72.6 |
| | ≥ 30 | 7 | 6 |
| Marital status | Married | 114 | 97.4 |
| | Widowed | 3 | 2.6 |
| Educational status | Informal education | 7 | 6 |
| | Primary (1–8) | 59 | 50.4 |
| | Secondary (9–12) | 32 | 27.4 |
| | Higher education | 19 | 16.2 |
| Occupational status | Gov't employee | 40 | 34.2 |
| | Unemployed | 15 | 12.8 |
| | Farmer | 3 | 2.6 |
| | Merchant | 44 | 37.6 |
| | House wife | 15 | 12.8 |
| Religion | Orthodox | 61 | 52.1 |
| | Muslim | 21 | 17.9 |
| | Protestant | 35 | 29.9 |
| Ethnicity | Oromo | 6 | 5.1 |
| | Amhara | 45 | 38.5 |
| | Tigre | 10 | 8.5 |
| | Bench | 18 | 15.4 |
| | Silte | 30 | 25 |
| | Others | 8 | 7.6 |
| Social drug use | Alcohol | 7 | 6 |
| | Chat | 7 | 6 |
| | No | 103 | 88 |

complications, with nephropathy making the highest proportion, 12 (44.4%). Among 72 (61.5%) of the participants who had comorbidities, 54 (75%) had only one comorbid condition, of which hypertension was the most common comorbid condition, 33 (28.2%), observed in T2DM patients. More than half of the participants, 61 (52.1%), had been on pharmacologic therapy for more than five years, and 11 (9.4%) of the study subjects visited the emergency department at least once a year. On the other hand, 44 (37.6%) of the participants took more than five medications per day, in which most of them were prescribed twice daily (88.9%).

## Factors associated with the occurrence of drug therapy problems

In bivariate logistic regression analysis, occupational status, number of comorbidities, and frequency of medications taken per day were candidates for multivariate analysis ($p \leq 0.25$). Nevertheless, in multivariate logistic regression, only occupational status and number of comorbidities were significantly associated with the occurrence of antidiabetic drug therapy

**Table 2. Drug therapy problems identified among adult T2DM patients on follow up at MTUTH, 2022.**

| Variable | Category | Frequency | Percent (%) |
|---|---|---|---|
| Need additional drug therapy | Yes | 50 | 42.7 |
| | No | 67 | 57.3 |
| Unnecessary drug therapy | Yes | 18 | 15.4 |
| | No | 99 | 84.6 |
| Ineffective drug therapy | Yes | 25 | 21.4 |
| | No | 92 | 78.6 |
| Dose too low | Yes | 13 | 11.1 |
| | No | 104 | 88.9 |
| Adverse drug reactions | Yes | 5 | 4.3 |
| | No | 112 | 95.7 |
| Dose too high | Yes | 16 | 13.7 |
| | No | 101 | 86.3 |
| Non-compliance | Yes | 45 | 38.5 |
| | No | 72 | 61.5 |
| Drug therapy problem | Yes | 83 | 70.9 |
| | No | 34 | 29.1 |

problems (DTPs) with a p-value of 0.03 and 0.02, respectively. Farmers were approximately four times more likely to develop DTPs than housewives (AOR = 3.56, 95% CI:1.12–11.38, P = 0.03). The odds of drug therapy problems were twice as high in those with four comorbid conditions than in those without comorbidities (AOR = 1.94, 95% CI:0.9–3.76, p = 0.02) (Table 4).

## Discussion

Among the 117 study participants in the current investigation, 172 drug therapy problems (DTPs) were identified, with an average of 1.47 DTPs per patient. This finding is comparable with the reports from Tikur Anbessa Specialized Hospital (TASH) and Oromia (Ethiopia), which showed an average of 1.16 ± 0.42 and 1.38 ± 0.85 DTPs per patient, respectively [12, 22]. However, it is higher than the findings from Sri Lanka, eastern Ethiopia, and Nigeria, which were found to be approximately 0.38, 0.86, and 1 DTPs per patient, respectively [24–26], and lower than the studies done in Harar [12], Wolaita Sodo [17], and Madda Walabu [16]. In contrast, the current study revealed that 83 (70.9%) participants experienced at least one drug therapy problem. This finding is higher than those reported from Hiwot Fana Specialized University Hospital (Harar, Eastern Ethiopia) (64.5%) [25] and Bahir Dar (62.4%) [11]. On the contrary, this is lower than the findings from Mettu (84.5%) [12], Addis Ababa (84%) [9], Wolaita Sodo (83.1%) [17], Jordan (81.2%) [13], Malaysia (90.5%) [8] and (91.8%) [27], and Madda Walabu (88%) [16]. These differences in the magnitude of DTPs across studies could be attributed to variability in the study settings, specific comorbidities considered, and the classification methods employed to classify DTPs. The majority of the studies sampled type 2 diabetes patients with comorbidities, most frequently hypertension, which increases the number of drugs the patients have to take, which in turn increases the likelihood of DTPs. The Cipolle and Pharmaceutical Care Network Europe (PCNE) were the most commonly employed methods for classifying DTPs [8, 11, 27–29].

In the present study, among the seven classes of DTPs identified, need for additional drug therapy (42.7%) and non-compliance (38.5%) were the most common problems. This is in line with a study conducted at Wolaita Soddo University Teaching Hospital (Southern Ethiopia),

**Table 3. Clinical and drug therapy conditions of adult type 2 diabetic patients on follow up at MTUTH, 2022.**

| Variable | Category | Frequency | Percent (%) |
|---|---|---|---|
| Renal function test | Normal | 18 | 15.4 |
| | Impaired | 7 | 6 |
| | Not done | 92 | 78.6 |
| Liver function test | Normal | 11 | 9.4 |
| | Impaired | 3 | 2.6 |
| | Not done | 103 | 88 |
| Presence of DM complications | Nephropathy | 12 | 10.3 |
| | Retinopathy | 8 | 6.8 |
| | Neuropathy | 7 | 6 |
| | No | 90 | 76.9 |
| Numbers of complications | One | 15 | 55.5 |
| | Two | 8 | 29.6 |
| | ≥3 | 4 | 14.8 |
| Number of comorbidities | One | 54 | 46.2 |
| | Two | 7 | 6 |
| | Three | 9 | 7.7 |
| | Four | 2 | 1.7 |
| | No comorbidity | 45 | 38.5 |
| Type of comorbidity | Hypertension | 33 | 28.2 |
| | Asthma | 3 | 2.6 |
| | Dyspepsia | 16 | 13.7 |
| | Dyslipidemia | 15 | 12.8 |
| | Heart failure | 5 | 4.3 |
| Number of hospitalizations per year | One | 19 | 16.2 |
| | Two | 8 | 6.8 |
| | ≥three | 2 | 1.7 |
| | No | 88 | 75.2 |
| Durations of treatment in year | ≤1 | 19 | 16.2 |
| | 2–4 | 37 | 31.6 |
| | ≥5 | 61 | 52.1 |
| Number of emergency visit per year | One | 11 | 9.4 |
| | Two | 8 | 6.8 |
| | ≥three | 2 | 1.7 |
| | No | 96 | 82.1 |
| Durations since diagnosis in years | <1 | 15 | 12.8 |
| | 2–4 | 41 | 35 |
| | ≥5 | 61 | 52.1 |
| Frequency of drug taken per day | Two | 104 | 88.9 |
| | Three | 13 | 11.1 |
| Total number of medications taken per day | <5 | 73 | 62.4 |
| | ≥5 | 44 | 37.6 |

which reported that need additional drug therapy and non-compliance were the most frequent problems, 56.37% and 51.9%, respectively [17]. Similarly, a study conducted at Madda Walabu University Goba Referral Hospital (Southeast Ethiopia) reported that needs additional drug therapy was the most frequent (50.2%) hurdle, followed by non-compliance (21.7%) [16]. A recent study conducted at a referral hospital in the capital city of Ethiopia (Addis Ababa)

**Table 4. Factors associated with the occurrence of drug therapy problems (DTPs) among patients with type 2 diabetes on follow up at MTUTH, 2022.**

| Variable | Category | COR (95% CI) | P- value | AOR (95% CI) | P-value |
|---|---|---|---|---|---|
| Occupational status | Gov't employee | 2.3(0.61–9.08) | **0.21** | 2.9(0.65–12.96) | 0.16 |
| | Unemployed | 9.4(0.75–119) | **0.08** | 5.025(0.24–104.35) | 0.29 |
| | Farmer | 3.2(1.19–9) | **0.02** | 3.564(1.12–11.38) | **0.03**\* |
| | Merchant | 0.72(0.13–3.96) | 0.71 | 0.822(0.13–5.08) | 0.83 |
| | House wife | 1 | | 1 | |
| Number of comorbidity | One | 0.32(0.13–0.78) | **0.01** | 1.77(0.34–9.33) | 0.99 |
| | Two | 2.34(0.9–3.57) | 0.99 | 1.9(0.38–9.05) | 0.5 |
| | Three | 0.63(0.14–2.82) | 0.54 | 3.3(1.22–89) | 0.99 |
| | Four | 0.94(0.57–4.57 | **0.09** | 1.95(0.9–3.76) | **0.02**\* |
| | No comorbidity | 1 | | | |
| Frequency of drug taken per day | Two | 1 | | 1 | |
| | Three | 3.32(1.03–10.78) | **0.05** | 3.891(0.673–22.500) | 0.13 |

\* Indicates statistical significance at p≤0.05.

pointed out needs additional drug therapy (25.1%) as the third most common problem, preceded by dose too low (28%) and ineffective drug therapy (26.1%) [22]. This might be due to the fact that different complications and comorbidities that occur following type 2 diabetes needs additional drug treatment.

In the current study, dose too high, dose too low, and ADRs, which accounted for 13.7%, 11.1%, and 4.3%, respectively, were the least prevalent DTPs. This finding is consistent with the studies from Nigeria [21], Malaysia [8], and Ethiopia [17], but contradicts the reports from Benin [14] and UK & Saudi Arabia [29], which disclosed ineffective drug therapy and adverse drug reactions as the most common DTPs. This may be because the latter studies included diabetic individuals with cardiovascular comorbidities, which raises the potential for drug-drug and drug-food interactions owing to the greater variety of medications the patients take. Moreover, most studies in Ethiopia that assessed drug-related problems in chronic patients reported adverse drug reactions among the least frequent problems. This may result from the failure of patients or healthcare professionals to recognize and/or report the issue.

This study identified that the number of comorbidities per patient was significantly associated with the occurrence of DTPs. Patients with four comorbidities were 1.9 times more likely to develop DTP than those without comorbidity. This finding is concordant with the studies conducted in UK & Saudi Arabia (SA), which revealed diabetic patients from the UK and SA with hypertension comorbidity were 1.25 and 3.99 times more likely to have DTPs than those without hypertension, respectively [29]. Similar findings are reported from Malaysia [8], Southern [17], and Central Ethiopia [22]. As explained previously, this might be due to the multiple/complex drug administration schedule, which contributes to non-compliance, adverse drug reactions, and cardiovascular risks that necessitate additional drug therapy.

This study also found that occupational status was significantly associated with the outcome variable. Being a farmer increased the odds of developing DTPs four-fold compared with homemakers. This result was nearly comparable with a study conducted on drug therapy problems among type 2 diabetes mellitus patients with hypertension in Ethiopia, which showed the relationship of occupational status with the occurrence of DTP in bivariate regression [11].

This may be explained by the fact that farmers tend to be busier and spend more time away from home, which may cause them to forget or skip their prescriptions.

## Limitation of the study

This study used a cross-sectional design that does not investigate a cause-and-effect relationship between the risk factors and the outcome variable. Being a retrospective study, identification of drug related problems mainly relied on patients' medical record; thus, presence of unrecorded data could affect the results of the study. The study did not address/intervene the identified DTPs. Selection bias is also possible, as the study sampled patients who were on follow-up visit during the study period.

## Conclusion and recommendation

In the present study, the proportion of type 2 diabetes patients with drug therapy problems was high. Need for additional drug therapy and non-compliance to medication were the two most common drug therapy problems identified. Occupational status and comorbidity were significantly associated with DTP occurrence. Undoubtedly, the presence of DTPs causes uncontrolled glycemia and the development of comorbid conditions, which increases the rate of morbidity and mortality. To reduce drug therapy issues and improve treatment outcomes of diabetic patients, it is essential to involve healthcare professionals with varied specialties, including skilled clinical pharmacists, in managing diabetic patients, which is a problem in virtually all hospitals in Ethiopia.

## Supporting information

**S1 File. Data collection tool/questionnaire.**
(DOCX)

**S1 Checklist. STROBE statement—Checklist of items that should be included in reports of observational studies.**
(DOCX)

## Acknowledgments

We would like to express our deepest gratitude to all participants in this research and to Dr. Morisky for allowing us to use the MMAS-8 (Morisky Medication Adherence Scale 8 item) in this research. We are also very grateful to all our beloved families for their unlimited support and motivation throughout the research.

## Author Contributions

**Conceptualization:** Semere Welday Kahssay, Nebeyi Fisseha Demeke.

**Data curation:** Semere Welday Kahssay, Nebeyi Fisseha Demeke.

**Formal analysis:** Semere Welday Kahssay.

**Investigation:** Semere Welday Kahssay, Nebeyi Fisseha Demeke.

**Methodology:** Semere Welday Kahssay, Nebeyi Fisseha Demeke.

**Project administration:** Semere Welday Kahssay, Nebeyi Fisseha Demeke.

**Resources:** Semere Welday Kahssay, Nebeyi Fisseha Demeke.

**Software:** Semere Welday Kahssay, Nebeyi Fisseha Demeke.

**Supervision:** Semere Welday Kahssay, Nebeyi Fisseha Demeke.

**Validation:** Semere Welday Kahssay, Nebeyi Fisseha Demeke.

**Visualization:** Semere Welday Kahssay, Nebeyi Fisseha Demeke.

**Writing – original draft:** Semere Welday Kahssay, Nebeyi Fisseha Demeke.

**Writing – review & editing:** Semere Welday Kahssay.

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
