## [Decision Letter · Decision Letter 0]

31 May 2023

PONE-D-23-09993Pharmacotherapy Problems and Associated Factors among Type 2 Adult Diabetic Patients on Follow up at Mizan-Tepi University Teaching Hospital, Southwest EthiopiaPLOS ONE

Dear Dr. Welday Kahssay

Thank you for submitting your manuscript to PLOS ONE. After careful consideration, we feel that it has merit but does not fully meet PLOS ONE’s publication criteria as it currently stands. Therefore, we invite you to submit a revised version of the manuscript that addresses the points raised during the review process.

We look forward to receiving your revised manuscript.

Kind regards,

Anmar Al-Taie, Ph.D.

Academic Editor

PLOS ONE

Journal Requirements:

- https://ijppr.humanjournals.com/wp-content/uploads/2018/08/15.Atluri-Bhavana-D.-S.-Kiran-A.-Kavitha-G.-Visweswara-Rao.pdf

- https://pubmed.ncbi.nlm.nih.gov/33029350/

- https://globaljournals.org/GJMR_Volume14/E-Journal_GJMR_(B)_Vol_14_Issue_7.pdf

- 

In your revision ensure you cite all your sources (including your own works), and quote or rephrase any duplicated text outside the methods section. Further consideration is dependent on these concerns being addressed.

Additional Editor Comments:

Comments for the authors

Abstract- Conclusion

• Consider removing any values for the study results from this part. This also applied for the conclusion part at the end of the manuscript.

Introduction

• Page 3, Line 48:

Consider writing (20 and 79 years old).

• Page 3, Line 56-57: ‘pharmacologic management of hypertension and hyperlipidemia’

Consider writing ‘pharmacologic management of comorbid disease conditions, such as hypertension and hyperlipidemia’.

Discussion

• Page 15, Line 231/232:

The findings do not match the order of the references.

• Page 15, Line 243-245:

Consider providing a reference for the sentence.

Limitations of the study

• More limitations are required to be reported for this study.

Reviewers' comments:

Reviewer's Responses to Questions

**Comments to the Author**

1. Is the manuscript technically sound, and do the data support the conclusions?

Reviewer #1: Yes

Reviewer #2: Yes

2. Has the statistical analysis been performed appropriately and rigorously? 

Reviewer #1: Yes

Reviewer #2: Yes

3. Have the authors made all data underlying the findings in their manuscript fully available?

Reviewer #1: Yes

Reviewer #2: Yes

4. Is the manuscript presented in an intelligible fashion and written in standard English?

Reviewer #1: Yes

Reviewer #2: Yes

5. Review Comments to the Author

Reviewer #1: Dear Author,

1) It is mentioned that “severely ill patients were excluded”. What do you want to mean as severely ill patients? What is its coverage? Could you define clearly?

2) Who and how many people collect the data? Who did provide training to data collectors? Can you define the concept of this training?

Reviewer #2: ‎ Dear Authors,‎

I enjoyed reading your manuscript. It is well-written and flows smoothly. However, I ‎have a few minor concerns that, if addressed, would greatly enrich your manuscript. Please find ‎my specific comments and suggestions below:‎

‎1. Please consider using the phrase "Patients with type II diabetes mellitus" instead of "Type 2 ‎Diabetic Patients."‎

In the study abstract:‎

‎2. I recommend modifying your objectives as follows:‎

Objectives: This study aimed to identify the incidence of drug therapy problems and determine ‎the associated factors among patients with type II Diabetes Mellitus at a University Teaching ‎Hospital in Southwest Ethiopia.‎

Regarding the Results section of the study abstract:‎

‎3. Please write "adjusted odds ratio" in full before using the abbreviation.‎

‎4. When reporting numerical values, limit the decimal places to two digits.‎

‎5. In your study results, you mentioned the following finding: "Farmers were about four times ‎more likely to develop DTPs than housewives (AOR=3.564, 95% CI: 1.116-11.384, P=0.03)." ‎Could you please elaborate on the clinical significance of this finding? Is it related to non-‎compliance with home medications?‎

In the Methodology section:‎

‎6. Provide further details about the study design, including the process of recruiting your study ‎sample and how participants were selected and approached.‎

‎7. Specify the ambulatory setting where you conducted your sample selection during follow-up ‎visits. Did you target endocrinology or family medicine clinics? I suggest describing the ‎ambulatory setting of the hospital and the specialties involved.‎

‎8. When providing operational definitions, make sure to cite all the references used after each ‎definition.‎

Regarding the Data Collection Instrument:‎

‎9. You mentioned using "A structured questionnaire prepared from previous studies with minor ‎modifications." Please cite the articles you used to prepare your questionnaire and describe the ‎specific modifications made.‎

‎10. Instead of describing the process of data collection, provide a detailed description of the ‎questionnaire used.‎

‎11. Create a separate section for the data collection process.‎

‎12. Clarify how training of data collectors took place. ‎

I have a few additional questions regarding the data collection process:‎

‎13. I didn't fully understand the reason for interviewing patients to detect DRPs. Could you ‎please describe the specific components of information retrieved directly from the patients' ‎medical files and those obtained during the interviews?‎

‎14. How did the interviews take place? Were they conducted in the clinic with the attending ‎physician or prior to entering the clinic?‎

‎15. In case the interviewers detected any DRPs, how did they respond? Did they inform the ‎prescriber or intervene to resolve these drug-related problems?‎

In the Results section:‎

‎16. Avoid starting a new sentence with numbers.‎

‎17. Please ensure consistency when reporting frequencies and percentages. For example, write ‎‎"27 (23.1%) of the study subjects developed microvascular complications, with nephropathy ‎accounting for the highest proportion." Remember to add a space between the frequency and the ‎percentage.‎

‎18. Add a footer to Table 4.‎

Lastly, it would be beneficial to mention any study limitations other than the issue of causality. ‎‎(e.g. selection bias as you only selected patients during a follow-up visits) ‎

Overall, your manuscript is well-structured and engaging. By addressing these minor concerns ‎and incorporating the suggested revisions, your work will be even more comprehensive and ‎accessible to the readers.‎

6. PLOS authors have the option to publish the peer review history of their article (what does this mean?). If published, this will include your full peer review and any attached files.

Reviewer #1: No

Reviewer #2: **Yes: **Rania Itani

---

## [Author Response · Author response to Decision Letter 0]

16 Jun 2023

Response to the editor

1. Journal requirements

Thank you. All the journal related issues raised (PLOS ONE’s style requirements, the minor overlapping issue, and caption) are addressed in the revised manuscript.

2. Abstract- Conclusion

• Consider removing any values for the study results from this part. This also applied for the conclusion part at the end of the manuscript.

Thank you for your comment. The figures/values are now removed from the conclusion parts (Please see line numbers 32-36 and 292-300 of the revised manuscript).

3. Page 3, Line 48: Consider writing (20 and 79 years old).

Thank you. The correction has been made (please see line number 47 of the revised manuscript)

4. Page 3, Line 56-57: ‘pharmacologic management of hypertension and hyperlipidemia’

Consider writing ‘pharmacologic management of comorbid disease conditions, such as hypertension and hyperlipidemia’.

Thank you for your comment. The correction has been made (please see line numbers 55 & 56 of the revised manuscript)

5. Discussion

• Page 15, Line 231/232: The findings do not match the order of the references

Thank you very much. The references are now correctly reordered (please see line numbers 232 & 233 of the revised manuscript).

6. Page 15, Line 243-245: Consider providing a reference for the sentence.

Thank you for your suggestion. References are now added (please see line numbers 240-246 of the revised manuscript).

7. Limitations of the study. • More limitations are required to be reported for this study.

Thank you. Limitations of the study section is now revised (please see line numbers 286-291 of the revised manuscript)

Response to Reviewer #1

Dear Reviewer #1, thank you for your constructive questions.

1. It is mentioned that “severely ill patients were excluded”. What do you want to mean as severely ill patients? What is its coverage? Could you define clearly?

Thank you for your concern. Severely ill patients refer those who were unable to respond to the questions due to their medical condition/s. It is now clearly stated in the revised manuscript (please see line numbers 96-97 of the revised manuscript).

2. Who and how many people collected the data? Who did provide training to data collectors? Can you define the concept of this training?

Thank you. Two pharmacists collected the data, and the principal investigator gave the training. Because the data collectors are experienced in data collection, the training mainly focused on explaining the components of the questionnaire. (Please see line number 148-149 & 153-154 of the revised manuscript)

Response to Reviewer #2

Dear Reviewer #2, thank you for your invaluable comments and suggestions.

1. Please consider using the phrase "Patients with type II diabetes mellitus" instead of "Type 2 ‎Diabetic Patients."‎

Thank you for your suggestion. Amendments are now made in the revised manuscript.

2. In the study abstract:‎ I recommend modifying your objectives as follows:‎

Objectives: This study aimed to identify the incidence of drug therapy problems and determine ‎the associated factors among patients with type II Diabetes Mellitus at a University Teaching ‎Hospital in Southwest Ethiopia.

Thank you very much for your recommendation. The correction has been made in the revised manuscript (please see line 14-16 of the revised manuscript)

Regarding the Results section of the study abstract:‎

3. ‎Please write "adjusted odds ratio" in full before using the abbreviation.‎

Thank you. The correction is now made. (Please see line number 29 of the revised manuscript). 

4. When reporting numerical values, limit the decimal places to two digits.‎

Thank you. The decimal places of all numerical values are limited to two digits. Please see the revised manuscript.

5. ‎In your study results, you mentioned the following finding: "Farmers were about four times ‎more likely to develop DTPs than housewives (AOR=3.564, 95% CI: 1.116-11.384, P=0.03)." ‎Could you please elaborate on the clinical significance of this finding? Is it related to non-‎compliance with home medications?‎

Thank you. Yes, it is attributed to non-compliance to prescribed medications. It is discussed in the discussion section, page 17, line 278-284 of the revised manuscript).

In the Methodology section:‎

6. Provide further details about the study design, including the process of recruiting your study ‎sample and how participants were selected and approached.‎

Thank you. Hospital-based cross sectional study design was used (line 83), and the participants were selected using a simple random sampling technique (line 117). Details on the process of approaching and recruiting the study subjects is explained in the ‘data collection method’ section (line 137)

7. ‎Specify the ambulatory setting where you conducted your sample selection during follow-up ‎visits. Did you target endocrinology or family medicine clinics? I suggest describing the ‎ambulatory setting of the hospital and the specialties involved.‎

Thank you. The ambulatory setting of MTUTH where diabetes patients receive a follow-up care is called diabetes outpatient clinic, and there are no specialties involved. (The correction has been made in the revised manuscript, line 86-87)

8. ‎When providing operational definitions, make sure to cite all the references used after each ‎definition.‎

Thank you for your suggestion. All operational definitions are now are cited. (Please see line numbers 119-136 of the revised manuscript).

Regarding the Data Collection Instrument:‎

9. ‎You mentioned using "A structured questionnaire prepared from previous studies with minor ‎modifications." Please cite the articles you used to prepare your questionnaire and describe the ‎specific modifications made.‎

Thank you for your constructive comment. A comprehensive questionnaire was prepared, and the minor modifications are on socio-demographic characteristics of the study subjects, in which some essential variables like social drug use and BMI (body mass index) were added. The references used are now cited. (Please see line 14 and 142-143 of the revised manuscript) 

10. ‎Instead of describing the process of data collection, provide a detailed description of the ‎questionnaire used.‎

Thank you. The contents of the questionnaire are sufficiently described (line numbers 139-152) and the tool is attached as a supplementary material.

11. ‎Create a separate section for the data collection process.‎

Thank you. All relevant information on the data collection process is stated in the ‘data collection tool and process’ section. (Please see line numbers 139-152 of the revised manuscript)

12. ‎Clarify how training of data collectors took place. ‎

Thank you. Training of the data collectors was carried out before initiation of the study. Because the data collectors are experienced in data collection, the training mainly focused on explaining the components of the questionnaire/data collection tool. Please see line numbers 155-156 of the revised manuscript.

 I have a few additional questions regarding the data collection process:‎

13. ‎I didn't fully understand the reason for interviewing patients to detect DRPs. Could you ‎please describe the specific components of information retrieved directly from the patients' ‎medical files and those obtained during the interviews?‎

Thank you. Patient interview focused on collecting sociodemographic characteristics of the study participants and assessing unreported adverse drug reaction, while clinical and medication-related information were collected retrospectively from the patients’ medical records. (Please see line numbers 144-146 of the revised manuscript)

14. ‎How did the interviews take place? Were they conducted in the clinic with the attending ‎physician or prior to entering the clinic?‎

Thank you. The study participants were interviewed before they enter into the diabetes outpatient clinic. (It is now included in the revised manuscript, line numbers 147 & 148)

15. ‎In case the interviewers detected any DRPs, how did they respond? Did they inform the ‎prescriber or intervene to resolve these drug-related problems?‎

Thank you. DRPs were detected mainly by reviewing their medical record. The study did not address/intervene the identified DRPs, which is included in the limitation of the study.

In the Results section:‎

16. ‎Avoid starting a new sentence with numbers.‎

Thank you, and your comment is considered in the revised manuscript.

17. ‎Please ensure consistency when reporting frequencies and percentages. For example, write ‎‎"27 (23.1%) of the study subjects developed microvascular complications, with nephropathy ‎accounting for the highest proportion." Remember to add a space between the frequency and the ‎percentage.‎

Thank you very much for the comment. Consistencies are now ensured in the revised manuscript.

18. ‎Add a footer to Table 4.‎

Thank you, and a footer in now added to Table 4 of the revised manuscript (please see line number 229 of the revised manuscript.

Lastly, it would be beneficial to mention any study limitations other than the issue of causality. ‎‎(e.g. selection bias as you only selected patients during a follow-up visits) .

Thank you. ‘Limitations of the study’ section is now revised (please see line numbers 286-291 of the revised manuscript).

---

## [Decision Letter · Decision Letter 1]

19 Jun 2023

Pharmacotherapy Problems and Associated Factors among Type 2 Adult Diabetic Patients on Follow up at Mizan-Tepi University Teaching Hospital, Southwest Ethiopia

PONE-D-23-09993R1

Dear Dr. Welday Kahssay

We’re pleased to inform you that your manuscript has been judged scientifically suitable for publication and will be formally accepted for publication once it meets all outstanding technical requirements.

Kind regards,

Anmar Al-Taie, Ph.D.

Academic Editor

PLOS ONE

Additional Editor Comments (optional):

All comments have been addressed

Reviewers' comments:

Reviewer's Responses to Questions

**Comments to the Author**

1. If the authors have adequately addressed your comments raised in a previous round of review and you feel that this manuscript is now acceptable for publication, you may indicate that here to bypass the “Comments to the Author” section, enter your conflict of interest statement in the “Confidential to Editor” section, and submit your "Accept" recommendation.

Reviewer #1: All comments have been addressed

Reviewer #2: All comments have been addressed

2. Is the manuscript technically sound, and do the data support the conclusions?

Reviewer #1: Yes

Reviewer #2: Yes

3. Has the statistical analysis been performed appropriately and rigorously? 

Reviewer #1: Yes

Reviewer #2: Yes

4. Have the authors made all data underlying the findings in their manuscript fully available?

Reviewer #1: Yes

Reviewer #2: Yes

5. Is the manuscript presented in an intelligible fashion and written in standard English?

Reviewer #1: Yes

Reviewer #2: Yes

6. Review Comments to the Author

Reviewer #1: The manuscript is technically sound, and the data supports the conclusions. Statistical analysis has been performed appropriately. I have no additional comment.

Reviewer #2: (No Response)

7. PLOS authors have the option to publish the peer review history of their article (what does this mean?). If published, this will include your full peer review and any attached files.

Reviewer #1: No

Reviewer #2: **Yes: **Rania Itani

---

## [Editor Report · Acceptance letter]

26 Jul 2023

PONE-D-23-09993R1 

Pharmacotherapy Problems and Associated Factors among Type 2 Adult Diabetic Patients on Follow up at Mizan-Tepi University Teaching Hospital, Southwest Ethiopia 

Dear Dr. Welday Kahssay:

I'm pleased to inform you that your manuscript has been deemed suitable for publication in PLOS ONE. Congratulations! Your manuscript is now with our production department. 

Kind regards, 

on behalf of

Dr. Anmar Al-Taie 

Academic Editor

PLOS ONE